# OpenReview forum: "Is Vibe Coding Safe? Benchmarking Vulnerability of Agent Generated Code in Real-World Tasks"
_ICLR.cc/2026/Conference — Submitted to ICLR 2026_

### Official Review · Reviewer_b85Q · 2025-10-26

**Soundness:** 2
**Presentation:** 2
**Contribution:** 3
**Rating:** 4
**Confidence:** 3

**Summary:**

This paper proposes SusVibes, a benchmark consisting of 200 software engineering tasks derived from real-world open-source projects, designed to evaluate the functionality and security capabilities of coding agents in vibe coding. In addition, the paper introduces an automatic curation pipeline that constructs repository-level coding tasks equipped with a runtime evaluation environment for end-to-end assessment.

**Strengths:**

1. Scope: The benchmark focuses on large-scale repositories with complex file structures, emphasizing realistic, repository-level coding environments.

2. Evaluation dimensions: It jointly assesses functional correctness and security correctness, ensuring that models are tested not only for task completion but also for safe and secure implementation.

3. Coverage: The dataset spans 77 CWE categories, capturing diverse vulnerability types across real-world projects.

4. Design intuition: The ideas behind Constraint I (Patch Enclosure) and Constraint II (Security Implications) are well-motivated and provide an intuitive foundation for constructing security-aware tasks.

5. Findings: Coding agents often achieve functional correctness but fail security checks on some tasks. Simple mitigation strategies—such as security-themed prompting, CWE self-identification, or even oracle CWE hints—do not reliably close this gap and often lead to a trade-off between functionality and security.

**Weaknesses:**

1. Unclear source of test cases:
The paper does not clearly specify where the functional and security test cases originate from—whether they are collected from real project test suites, generated by LLMs, or synthesized during the curation pipeline.

2. Pipeline clarity and reproducibility:
While the three-stage pipeline (masking, problem generation, verification) is novel and interesting, its description remains conceptually vague. The paper lacks concrete examples or visual case studies showing how a real repository sample passes through each stage.

3. Quality of generated masks:
It is difficult to ensure the mask quality, particularly whether masking removes or alters critical contextual information needed for code understanding and security reasoning.

4. Inaccuracy in mask ratio control:
The mask ratio (α) is controlled by the Stage-1 agent, which might not consistently or accurately capture the right functional boundary—especially across large repositories with complex dependencies.

5. CWE scope ambiguity:
The paper does not make clear whether each task corresponds to a single CWE or involves multiple overlapping CWE types.

**Questions:**

1. Where do the functional and security test cases come from?

2. Can the authors provide a concrete case study demonstrating the three-stage pipeline process (Stage 1–3) on a real vulnerability example?

3. Does each task map to a single CWE, or can it belong to multiple CWEs simultaneously?

3. How is mask-ratio control validated? Has any quantitative analysis been done to verify that the Stage-1 agent accurately identifies proper masking boundaries?

---

> ### Author Response · Authors · 2025-11-25
>
> We thank the reviewer for the thoughtful review and constructive suggestions. We appreciate that the reviewer recognizes SusVibes’ focus on large, realistic repositories and its joint evaluation of functionality and security, and regards the benchmark design as a useful step toward assessing secure vibe coding in practice. We would like to address the reviewer’s concerns as below:
>
> **W1. Unclear source of test cases & Q1. Where do the functional and security test cases come from?**
>
> A1: Thanks for allowing us to clarify this question. The functionality and security tests used in our benchmark are sourced from human-written test suites of real-world GitHub repositories. The detailed process of how we obtain potential security tests and functionality tests are as follows:
>
> (1) The task curation starts at a commit that fixes a known vulnerability in the implementation of an existing feature $\mathcal{F}$ with the patch $\mathcal{P}$. We filter out the commits that do not modify the test suite, because those would not contain security tests that can detect the fixed vulnerabilities. For a single vulnerability fixing commit, we separate the changes it made P into two parts: the one that modifies the implementation of $\mathcal{F}$, and the one that modifies the test suite. The added tests from the vulnerability fixing commits are collected as potential security tests, and they can be added to the repository by applying the patch that changes the test suite.
>
> (2) After running security tests, we check out the code base back to the commit before the fix, which contains the vulnerable implementation of the feature $\mathcal{F}$, and the corresponding functionality tests.
>
> The above pipeline provides sources of human-written tests that are likely to examine functional correctness and security of the feature implementation.
>
> We further rigorously validate these tests based on execution results. We run different combinations of feature implementations and test suites in the execution environment, and check if the following requirements are satisfied: (i) the masked vulnerable commit must fail both functional and secure tests; (ii) the code base with vulnerable implementation needs to pass functional tests but fail secure tests; and (iii) the vulnerability fix commit needs to pass both test cases. Through this validation, we obtain functionality and security tests that are used in evaluation.
>
> ---
>
> **W2. Pipeline clarity and reproducibility: The paper lacks concrete examples or visual case studies showing how a real repository sample passes through each stage. Q2. Can the authors provide a concrete case study demonstrating the three-stage pipeline process (Stage 1–3) on a real vulnerability example?**
>
> A2: We have organized our writing with a concrete example showcasing the three-stage pipeline curation process in our revised manuscript. The complete description can be found from lines #192 to #247, and demonstrated in Figure 2 in our revision.
>
> To summarize:
>
> The task curation starts at a vulnerability fixing commit $\mathcal{C}_0$ from the GitHub repository django. It fixes a known vulnerability in an existing feature implementation $\mathcal{F}$ (`verify_password()`). It fixes several lines within the `verify_password()`function to ensure that under all input cases, the function takes roughly the same time to finish (a detailed explanation of why this is a security concern is described in Section 4.3 of our manuscript).
>
> (i) The code base is then checkout back to the commit before the fix, where we utilize an LLM agent to mask out the feature $\mathcal{F}$, i.e. the whole function `verify_password()`, from its vulnerable implementation, obtaining $\mathcal{C}_{-1}^{\mathcal{M}}$, the repository without $\mathcal{F}$. From this code base version, we create a task of requesting the full `verify_password()` function back to the code base.
>
> (ii) We generate a task description for re-implementing the feature $\mathcal{F}$ `verify_password()` based on its vulnerable implementation that was masked out, describing its functionality.
>
> (iii) We employed a verifier agent to check whether the generated task description accounts for all lines in the implementation of the feature `verify_password()` with security fixes. As detailed in Figure 3 of our revision, the agent is prompted to match each line in the secure implementation to a requirement in the description. If any line in the secure implementation is not mentioned by the task description, it will go back to step (i) to regenerate a larger mask; otherwise, the curation pipeline successfully produces the task description and the mask that matches the secure/golden implementation.

---

> ### Author Response · Authors · 2025-11-25
>
> **W3. Quality of generated masks & W4. Inaccuracy in mask ratio control & Q4. How is mask-ratio control validated?**
>
> A3: To ensure the mask can appropriately cover the feature implementation and the security fixes, we design an iterative masking pipeline with verification. It includes:
>
> (i) An agent to generate an initial mask $\mathcal{M}$. This mask $\mathcal{M}$ is generated on the vulnerable commit before the security fix, i.e., masking out a feature $\mathcal{F}$ from its vulnerable implementation.
>
> (ii) A task description is generated to describe the functionality of this masked implementation
>
> (iii) A verifier agent is used to check whether the task description covers all lines in feature implementation $\mathcal{F}$ + security fixes. Specifically, the agent is prompted to match each line in the secure implementation to a requirement in the description. If any line in the secure implementation is not mentioned by the task description, it will go back to step (i) to regenerate a larger mask, otherwise, go to step (iv)
>
> (iv) Return the task description and the mask that matches the secure/golden implementation
>
> To conclude, instead of solely relying on one agent to do verification, this curated pipeline can ensure that the masked feature and task description cover the feature implementation $\mathcal{F}$ + security fixes. This supervision can mitigate the blind spots of LLM agents in verification. We also elaborate on it more in our revision from line #192 to line #247.
>
> Furthermore, we conduct human evaluation on a randomly sampled subset of 15 tasks (7%) to manually verify the quality. These 15 tasks span across 12 different GitHub projects, accessing 14 different CWEs, forming a representative subset of SusVibes.
>
> 3 software engineering annotators assess the quality of the mask generated by the LLM based on the following criteria:
>
> 1. **Valid Mask:** the mask purely deletes an implementation (or adds at most some place holders)
> 2. **Sufficient Mask:** The mask removes a sufficient feature implementation surrounding the security fix, containing enough security context.
>     1. If the mask covers all lines that were touched by the security fix patch.
>     2. If the mask covers the implementation of a feature that requires the security fix.
> 3. **Necessary Mask:** The mask avoids obviously unrelated or excessive deletions.
>
> The resulting table is shown below.
>
> | Project | CWE Examined | Lines Deleted | Mask is Valid | Mask is Sufficient | Mask is Necessary |
> | --- | --- | --- | --- | --- | --- |
> | `aiohttp` | CWE-444 | 63 | ✓ | ✓ | ✓ |
> | `airflow` | CWE-78 | 83 | ✓ | ✓ | ✓ |
> | `airflow` | CWE-20 | 60 | ✓ | ✓ | ✓ |
> | `airflow` | CWE-79 | 36 | ✓ | ✓ | ✓ |
> | `ckan` | CWE-344 | 115 | ✓ | ✓ | ✓ |
> | `django` | CWE-770 | 225 | ✓ | ✓ | ✓ |
> | `django` | CWE-1333 | 176 | ✓ | ✓ | ✓ |
> | `pandas-ai` | NVD-CWE-noinfo | 151 | ✓ | ✓ | ✓ |
> | `rdiffweb` | CWE-1021 | 106 | ✓ | ✓ | ✓ |
> | `flask` | CWE-539 | 45 | ✓ | ✓ | ✓ |
> | `paramiko` | CWE-362 | 156 | ✓ | ✓ | ✓ |
> | `plone.namedfile` | CWE-79 | 349 | ✓ | ✓ | ✓ |
> | `salt` | CWE-22 | 150 | ✓ | ✓ | ✓ |
> | `vyper` | CWE-787 | 299 | ✓ | ✓ | ✓ |
> | `django-mfa3` | CWE-287 | 169 | ✓ | ✓ | ✓ |
> | *12 projects* | *14 CWEs* | *145.5 lines* | *100%* | *100%* | *100%* |
>
> It is suggested by human software engineers that the masks constructed from the generation-verification pipeline in SusVibes are reasonable in forming security-oriented coding tasks.
>
> ---
>
> **W5. CWE scope ambiguity & Q5. Does each task map to a single CWE, or can it belong to multiple CWEs simultaneously?**
>
> A1: We thank the reviewer for pointing this out and will report the CWE distribution more clearly in the revision. Most tasks in SusVibes (97.5%) correspond to a single CWE, while only 3.5% correspond to multiple CWEs, and the maximum number of CWEs each task corresponds to is 2. We will include summary statistics (average CWEs per task and the fraction of single- vs. multi-CWE tasks) and a short table in the appendix.

---

### Official Review · Reviewer_3D9P · 2025-10-26

**Soundness:** 4
**Presentation:** 2
**Contribution:** 3
**Rating:** 4
**Confidence:** 4

**Summary:**

This paper introduces SUSVIBES, a benchmark of 200 repository-level coding tasks derived from real-world security vulnerabilities, to evaluate the security of LLM-based coding agents. The study finds that while frontier models achieve reasonable functional correctness, the vast majority of their solutions contain security vulnerabilities, and simple prompting strategies fail to mitigate this issue while degrading functional performance.

**Strengths:**

Highly relevant and timely problem: SUSVIBES operates at the repository level, requiring cross-file edits and build/test interactions, a major step beyond existing single-file benchmarks, also, from the reviewer’s point of view, vibe coding is one of the timely topics, making this research critically important for the current AI-assisted programming landscape.

Novel benchmark construction: The three-stage automated pipeline systematically constructs tasks from real vulnerabilities, ensuring scalability and authenticity, which can also be extended to other applications

Good evaluation and insights: This work identifies strong evidence that current agents are functionally competent but insecure with case studies such as Django password verification examples, clearly demonstrates timing-based vulnerability reproduction by LLMs, providing the readers with a straightforward and clear idea of the task of benchmarking vibe coding on LLMs.

**Weaknesses:**

Data imbalance: One of the concerns is from Fig 3, among all 10 categories, the data distribution is imbalanced, which may affect the effectiveness of the evaluation with SUSVIBES.

Potential dataset bias: The benchmark heavily depends on public vulnerability repositories, which may favor well-documented projects and miss obscure bug classes.

Insufficient mitigation approach: This work has insights into how insecure the current vibe coding is, while proposing some potential mitigation approaches beyond prompt based techniques will make this paper more solid

Analysis can be improved: From the reviewer’s aspect, security correctness in this work is defined by test outcomes; weak or incomplete tests could underestimate latent vulnerabilities. Also, only a few models such as Claude, Gemini and Kimi are tested, which I think is not sufficient. How’s other model family, for example, Qwen, GPT etc. performs on these tasks.

**Questions:**

Can you please justify why swe-agent was chosen to generate the data? Have you tested other agent systems? How good are they?

Also back to the weakness on analysis, why did you choose Claude, Kimi and Gemini but not other models?

Are there plans to release a leaderboard or ongoing benchmark updates as new vulnerabilities or projects emerge?

How do you ensure the difficulty distribution across 200 tasks is reasonable? Could some tasks be too trivial or too difficult, affecting the benchmark's discriminative power?

Fig. 4 shows different models excel at different CWE types. I am curious if this work also analyzed which CWE types are inherently harder to detect or fix? What are the root causes of these differences?

What specific mitigation strategies or best practices would you recommend for engineering teams wanting to use vibe coding in production, as it was proposed in the weakness?

**Details Of Ethics Concerns:**

It would be better to discuss 1. How should people avoid abusing vulnerable code from Vibe Coding 2. Benchmark tasks are from real-world projects; are they properly cited or credited? Thus, I think it may have security and terms of use problems.

---

> ### Author Response · Authors · 2025-11-25
>
> We are grateful to the reviewer for the careful and insightful assessment. We are glad that the reviewer finds SUSVIBES a timely repository-level benchmark grounded in real vulnerabilities, and that the empirical analysis of the gap between functional correctness and security is viewed as informative for understanding the risks of vibe coding. We would like to address the reviewer’s concerns as below:
>
> **W1. Data imbalance: One of the concerns is from Fig. 3, among all 10 categories, the data distribution is imbalanced, which may affect the effectiveness of the evaluation with SUSVIBES**
>
> A1: We agree that a more balanced data distribution can be helpful, and extra human annotation should be done to annotate the security issue and the security test suite. However, the focus of this paper is to propose an automatic testbed for repository-level benchmarking on top of the existing human-annotated CWEs and highlight the security concern in Vibe Coding. We would like to leave the extra human annotations for a more balanced distribution as our future work.
>
> Besides, we would like to clarify that we do not manually select the security domains in SusVibes. Instead, we filter GitHub repositories based on the criteria (Python > 3.7, commit contains security test suite, reproducibility execution environment). The number in each security domain depends on how much data satisfies these criteria, which is also affected by the original repository distribution on GitHub and the distribution of real-world vulnerabilities. For example, ChatBots projects are less vulnerable in CWEs SusVibes care about, leading to only 2 projects, while in identity authorization or networking domains, they might be more vulnerable.
>
> Moreover, compared with previous work, our SusVibes shows a more diverse distribution on realistic application domains. **CWEval** consists of hand-crafted, function-level security-critical coding tasks organized by CWE type and programming language, but it does not model or report how vulnerabilities are distributed across application domains. **BaxBench** offers 28 author-designed small backend tasks (scenarios) across multiple frameworks, yet all tasks live within a single HTTP web-backend setting rather than spanning distinct domains. **Asleep** evaluates small code snippets around top CWEs, without tying tasks to concrete application domains. **SecCodePLT** starts from manually designed security tasks and scales them synthetically with LLM-based mutations, organizing the benchmark by CWE categories and task families instead of domains. These previous benchmarks restrict context to a single function or file, which limits their ability to capture realistic in-deployment application scenarios.
>
> ---
>
> **W2. Potential dataset bias: The benchmark heavily depends on public vulnerability repositories, which may favor well-documented projects and miss obscure bug classes.**
>
> A2: We thank the reviewer for raising this point and agree that our benchmark cannot cover all potential security vulnerabilities that Vibe coding may introduce. However, our benchmark provides a repository-level testbed for the security issue in vibe coding and covers more vulnerabilities than existing secure coding benchmarks, as shown in Table 1. SusVibes also covers vulnerabilities that cannot be classified into any existing categories (2% of the tasks), and demonstrates extendability that once novel, publicly recorded vulnerabilities are discovered, they can be easily adapted into the benchmark by tracing back to the vulnerable commit and synthesizing the feature request and the runtime evaluation environment.
>
> ---
>
> **W3. Insufficient mitigation approach: This work has insights into how insecure the current vibe coding is, while proposing some potential mitigation approaches beyond prompt-based techniques will make this paper more solid**
>
> A3: We appreciate the valuable suggestion that discussing mitigation strategies beyond prompt-based techniques can strengthen the paper. We would like to highlight that our main focus is to (i) construct a realistic, security-focused benchmark for vibe coding and (ii) demonstrate that the frontier software engineering agents have severe security risks and provide qualitative and quantitative analysis. We would like to leave the investigation of better mitigation methods as our future work.

---

> ### Author Response · Authors · 2025-11-25
>
> **W4. Analysis can be improved: How do other model family, for example, Qwen, GPT etc., performs on these tasks. & Q2.why did you choose Claude, Kimi, and Gemini but not other models?**
>
> A4: We thank the reviewer for the suggestion and we agree that the other LLMs, such as Qwen and GP,T may share the same security issue. We believe that our new benchmark, SusVibes, can contribute to the community by showing the security concerns in Vibe Coding and setting up a comprehensive testbed for Vibe Coding. The other model families such as Qwen and GPT, can also be easily evaluated in our SusVibes. In this paper, we choose Claude, Kimi, and Gemini because they are the frontier LLMs in agentic tasks and these cover both open-sourced and closed-sourced LLMs to demonstrate the generalization of our conclusions.
>
> Due to time and cost constraints, we are not able to cover all these LLMs in our paper. As agents may iterate over 100 steps to complete a single complex software engineering task, evaluation is time and cost-intensive, and the set of models and agents we evaluated is currently restricted. In total, we have spent more than $5000 in building SusVibes and testing the Claude, Gemini, and Kimi.
>
> We will be releasing the leaderboard, dataset, and evaluation submission portal. Once these are available to the community, we expect that additional model families (e.g., Qwen, GPT) and agent frameworks, as well as newly introduced methods, can be evaluated on SUSVIBES in a scalable, collaborative way. In this sense, the current experiments should be viewed as an initial baseline, and we explicitly position SUSVIBES as a shared benchmark on top of which the broader community can systematically compare and improve more models over time.
>
> ---
>
> **Q1. Can you please justify why swe-agent was chosen to generate the data? Have you tested other agent systems? How good are they?**
>
> A1: We employ the best performing agent on SWE-Bench, which may potentially have the highest capability in editing the codebase and completing the benchmark creation task.
>
> ---
>
> **Q3. Are there plans to release a leaderboard or ongoing benchmark updates as new vulnerabilities or projects emerge?**
>
> A3: We will be releasing a leaderboard, GitHub repo operating as a portal that allows the public community to submit evaluation results, as well as continuously updating the benchmark. We believe these can contribute to the community by showing the security concerns in Vibe Coding and setting up a comprehensive testbed for Vibe Coding.
>
> ---
>
> **Q4. How do you ensure the difficulty distribution across 200 tasks is reasonable? Could some tasks be too trivial or too difficult, affecting the benchmark's discriminative power?**
>
> A4. We demonstrate the difficulty of SusVibes tasks through the largely variational evaluation outcomes of these tasks across different models/agents. Our evaluation settings assess agent scaffolds: OpenHands, SWE-agent on top of frontier LLMs: Claude 4, Kimi K2, Gemini 2.5 Pro, resulting in 6 different combinations of models and agents. In SusVibes, while some tasks are easily solved by any agents, some are difficult to solve by a very limited number of agents, and the others are solved in between by a subset of agent settings.
>
> In terms of functionality difficulty, we calculate the set of tasks solved correctly in each agent setting. Specifically, we measure the number of agents that produce a correct solution for each task in SusVibes and report the distribution as follows.  Concretely, across the 200 tasks, 5.5% (11/200) are solved by all 6 agents, and the remaining 54.0% (108/200) are solved by between 1 and 5 agents (28, 27, 25, 15, and 13 tasks, respectively). Since the lower the number of agents, the harder the task, our finding is that in SusVibes, the percentage of tasks gradually increases as the difficulty level increases. 40.5% (81/200) of tasks are not solved by any of the 6 agent–model combinations, which provides a considerable potential for evaluating and distinguishing further fast-advancing agents and LLMs. This spread shows that SusVibes contains a substantial tail of hard tasks, a small set of relatively easy ones, and a majority of tasks with intermediate difficulty, providing good discriminative power when comparing different agents and models.
>
> | # of agents that solved the task | Tasks (fraction) |
> | --- | --- |
> | 6 | 0.055 |
> | 5 | 0.065 |
> | 4 | 0.075 |
> | 3 | 0.125 |
> | 2 | 0.135 |
> | 1 | 0.140 |
> | 0 | 0.405 |
>
>
>
>
>
>
>
> In terms of security, the trend of the difficulty of SusVibes tasks is similar. In detail, for example, although Claude can solve significantly more tasks correctly than Gemini (53% v.s. 16%), on the intersection of tasks solved correctly for both Claude 4 and Gemini, Claude only solves 50% of tasks securely solved by Gemini. Detailed variational security performance of each model and agent scaffold is shown in Figure 4 of the original manuscript.

---

> > ### Author Response · Authors · 2025-11-25
> >
> > **Q5. Fig. 4 shows different models excel at different CWE types. I am curious if this work also analyzed which CWE types are inherently harder to detect or fix? What are the root causes of these differences?**
> >
> > A5: Thank you for the insightful question. Our primary goal in this work is to compare models under the same set of repository-level tasks, so we are cautious about making strong claims about inherent CWE difficulty, since these observed success rates are also influenced by factors like codebase complexity, scenario, training of different models, etc. That said, our per-CWE performance breakdown does exhibit systematic differences to some extent, and we discuss an example of two different groups of CWEs below.
> >
> > Concretely, we find that
> >
> > - **pattern-like, locally-fixable vulnerabilities tend to be easier for models**. For example, CWEs related to input validation/injection (e.g., an XSS/SQLi-style issue where the fix is to apply an appropriate sanitizer/parameterization at a single call site) are often resolved correctly: the vulnerability is localized to a small region of code, and there are many similar “sanitize → propagate through the same call chain” patterns in the models’ pretraining data. In these cases, agents can succeed via relatively shallow reasoning plus pattern matching: trace the immediate dataflow from user input to a sink and insert the missing validation or API call.
> > - **cross-cutting or configuration-oriented CWEs are noticeably harder.** For instance, CWEs related to access control/permissions or authentication (e.g., fixing subtle authorization checks, permission masks, or TLS / header configuration) often require the agent to reason about global invariants (who should be allowed to call what, under which configuration flags), understand framework-specific conventions, and modify locations or configuration points in a consistent way. In our benchmark, models frequently appear to fix the obvious local symptom while missing other call paths or configuration knobs, or they introduce new inconsistencies (e.g., tightening a check in one code path but forgetting another entry point).
> >
> > In our experiments, the second group (CWE-285/284 for authorization and access control, CWE-306/287 for authentication) demonstrate to be harder than the first group of vulnerabilities (CWE-79/80 for input validation and injection, CWE-352 for cross-site scripting) consistently (e.g. the secure pass ratio for Claude 29% v.s. 20%; for Kimi 16% v.s. 0%). We suspect the root causes behind the scene may include:
> >
> > - **Locality of the fix.** CWEs whose fix is localized to one function or call site are easier than those requiring coordinated edits across handlers, middleware, configuration, and tests.
> > - **Context depth.** Some CWEs are almost locally decidable: for issues like XSS or SQL injection, whether the code is secure can often be inferred from a relatively small snippet where raw user input flows into HTML or a query. By contrast, missing authentication/authorization checks are highly context dependent—whether a particular endpoint “should” enforce an origin check or a permission guard depends on how routing, middleware, and security policies are structured across the repository—so the agent must inspect and reason over much larger portions of the codebase.
> > - **Framework and domain knowledge.** Some CWEs (e.g., around TLS headers, authentication middleware, or permission models) depend on framework-specific idioms that the model may not reliably recall or adapt across large repositories.
> >
> > ---

---

> > > ### Author Response · Authors · 2025-11-25
> > >
> > > **Q6. What specific mitigation strategies or best practices would you recommend for engineering teams wanting to use vibe coding in production, as was proposed in the weakness?**
> > >
> > > A6: Thank you for raising this practical point. Our work primarily focuses on measuring the security risks of vibe coding, rather than on proposing a mitigation framework, but our findings do suggest several concrete best practices.
> > >
> > > We would like to emphasize the importance of context (especially over large repositories) in implementing security practices. Our results indicate that insufficient use of repository context is a major failure mode: agents often patch the most visible local symptom without understanding existing security mechanisms (e.g., middleware, decorators, global configuration), which can lead either to incomplete fixes or to inconsistent policies. A central mitigation, therefore, is to explicitly induce agents to inspect the broader context before editing. In practice, this includes:
> > >
> > > 1. requiring the agent to perform and summarize a repository-wide search for relevant handlers, middleware, configuration, and connection points before proposing a patch; 、
> > > 2. asking the agent to explicitly state what security invariants it believes the system should enforce (e.g., which endpoints require authentication, which inputs must be validated) and where these are currently implemented; and
> > > 3. Enforcing a workflow where the agent first produces a “plan over context” (files to inspect, security checks to reuse) and only then generates code changes. These simple orchestration changes may make it more likely that the agent can have a better global understanding to be aware of the security risks and how to correctly defend against them.
> > >
> > > However, as we discussed in our mitigation approaches, a trade-off exists in current agent systems that once an agent is induced to spend more attention on one part of the task (e.g., context examination before implementing), it might put less effort into other parts (e.g., iteratively testing and refining the code it writes). A more promising direction is to factor these responsibilities into separate sessions and reduce the total amount of detailed information one LLM(agent) can see. Such modular, or potentially multi-agent designs can relieve the workload any single agent must handle at once while still combining global understanding with rigorous verification.
> > >
> > > **References:**
> > >
> > > [1] Asleep at the Keyboard? Assessing the Security of GitHub Copilot's Code Contributions
> > >
> > > [2] BaxBench: Can LLMs Generate Secure and Correct Backends?
> > >
> > > [3] CWEval: Outcome-driven Evaluation on Functionality and Security of LLM Code Generation
> > >
> > > [4] SecCodePLT: A Unified Platform for Evaluating the Security of Code GenAI

---

### Official Review · Reviewer_DRLX · 2025-11-01

**Soundness:** 2
**Presentation:** 3
**Contribution:** 2
**Rating:** 4
**Confidence:** 4

**Summary:**

This paper builds a repository-level secure code generation benchmark with an automatic pipeline.

**Strengths:**

- It extends secure code generation benchmarks to the repository level, which is more realistic and closer to real-world development scenarios.
- It provides an automatic pipeline for constructing tasks, which has potential to extend to more projects.

**Weaknesses:**

- The masking process is done by an LLM. Did the authors manually examine the masked results to ensure they are reasonable?
- In Section 3.1.1, the paper mentions 'a test patch T, likely examining security.' How does the automatic pipeline determine whether a test is related to security, even if it appears in the same commit as the patch?
- Since the entire problem is generated by LLM, how do the authors justify that each task is reasonable and not vague or missing critical information? For example, could a senior human software engineer complete the task and produce secure code with the provided context?
- The benchmark only includes 200 tasks, despite the authors claiming to have collected ~20,000 open-source vulnerability records over the last 10 years. This raises concerns about the success rate of the pipeline.

**Questions:**

- I understand the security issue and the need for a fake runtime in the Django example, but could the authors further explain how this example relates to 'spam emails or junk messages'?
- Could the authors provide more task examples (like Figure 6) or release the entire dataset? Having access to more examples would help better assess the quality of the benchmark.
- The authors claim the benchmark consists of 100% cross-file edits. Could they explain why the Django example is considered a cross-file edit, given that it appears to modify only one function in a single file?

---

> ### Author Response · Authors · 2025-11-25
>
> We thank the reviewer for the detailed and helpful comments and appreciate the recognition of our contributions. We admit and appreciate it that the reviewer values our effort to move secure code generation evaluation to realistic repository-level settings and to design an automatic pipeline that can scale to additional projects. We would like to address some concerns as below:
>
> **W1. The masking process is done by an LLM. Did the authors manually examine the masked results to ensure they are reasonable? & W3. How do the authors justify that each task is reasonable and not vague or missing critical information?**
>
> A1: To ensure the mask can appropriately cover the feature implementation and the security fixes, we design an iterative masking pipeline with verification. It includes:
>
> (i) An agent to generate an initial mask $\mathcal{M}$. This mask $\mathcal{M}$ is generated on the vulnerable commit before the security fix, i.e., masking out a feature $\mathcal{F}$ from its vulnerable implementation.
>
> (ii) A task description is generated to describe the functionality of this masked implementation
>
> (iii) A verifier agent is used to check whether the task description covers all lines in feature implementation $\mathcal{F}$ + security fixes. Specifically, the agent is prompted to match each line in the secure implementation to a requirement in the description. If any line in the secure implementation is not mentioned by the task description, it will go back to step (i) to regenerate a larger mask, otherwise, go to step (iv)
>
> (iv) Return the task description and the mask that matches the secure/golden implementation
>
> To conclude, instead of solely relying on one agent to do verification, this curated pipeline can ensure that the masked feature and task description cover the feature implementation $\mathcal{F}$ + security fixes. This supervision can mitigate the blind points of LLM agents in verification. We also elaborate on it more in our revision from line #192 to line #247.
>
> Furthermore, we conduct human evaluation on a randomly sampled subset of 15 tasks (7%) to manually verify the quality. These 15 tasks span across 12 different Github projects, accessing 14 different CWEs, forming a representative subset of SusVibes.
>
> 3 software engineering annotators assess the quality of the mask generated by the LLM based on the following criteria:
>
> 1. **Valid Mask:** the mask purely deletes an implementation (or adds at most some place holders)
> 2. **Sufficient Mask:** the mask removes a sufficient feature implementation surrounding the security fix, containing enough security context.
>     1. If the mask covers all lines which were touched by the security fix patch.
>     2. If the mask covers the implementation of a feature which requires the security fix.
> 3. **Necessary Mask:** The mask avoids obviously unrelated or excessive deletions.
>
> The resulting table is shown below.
>
> | Project | CWE Examined | Lines Deleted | Mask is Valid | Mask is Sufficient | Mask is Necessary |
> | --- | --- | --- | --- | --- | --- |
> | `aiohttp` | CWE-444 | 63 | ✓ | ✓ | ✓ |
> | `airflow` | CWE-78 | 83 | ✓ | ✓ | ✓ |
> | `airflow` | CWE-20 | 60 | ✓ | ✓ | ✓ |
> | `airflow` | CWE-79 | 36 | ✓ | ✓ | ✓ |
> | `ckan` | CWE-344 | 115 | ✓ | ✓ | ✓ |
> | `django` | CWE-770 | 225 | ✓ | ✓ | ✓ |
> | `django` | CWE-1333 | 176 | ✓ | ✓ | ✓ |
> | `pandas-ai` | NVD-CWE-noinfo | 151 | ✓ | ✓ | ✓ |
> | `rdiffweb` | CWE-1021 | 106 | ✓ | ✓ | ✓ |
> | `flask` | CWE-539 | 45 | ✓ | ✓ | ✓ |
> | `paramiko` | CWE-362 | 156 | ✓ | ✓ | ✓ |
> | `plone.namedfile` | CWE-79 | 349 | ✓ | ✓ | ✓ |
> | `salt` | CWE-22 | 150 | ✓ | ✓ | ✓ |
> | `vyper` | CWE-787 | 299 | ✓ | ✓ | ✓ |
> | `django-mfa3` | CWE-287 | 169 | ✓ | ✓ | ✓ |
> | *12 projects* | *14 CWEs* | *145.5 lines* | *100%* | *100%* | *100%* |
>
> It is suggested by human software engineers that the masks constructed from the generation-verification pipeline in SusVibes are reasonable in forming security-oriented coding tasks.
>
> ---
>
> **W2. How does the automatic pipeline determine whether a test is related to security, even if it appears in the same commit as the patch?**
>
> A2:  We thank the reviewer for pointing out this question. A secure test case should fail on the pre-commit code and succeed on the vulnerability-fix (post-commit) code. Therefore, we run the pre-commit code and vulnerability-fix code under our execution environment, and validate that each task corresponds to test cases that change from fail to pass as the security test cases. This execution-based validation is shown in our original manuscript line #236, we further improved the writing in our updated manuscript and illustrate this at line #260.

---

> ### Author Response · Authors · 2025-11-25
>
> **W4. The benchmark only includes 200 tasks, despite the authors claiming to have collected ~20,000 open-source vulnerability records over the last 10 years. This raises concerns about the success rate of the pipeline.**
>
> A4: We thank the reviewers for raising this point. Before we use our pipeline to curate the task, we first filter the vulnerability records based on these criteria (Lines #184 to #189 in our original manuscript and line #197 to #202 in our updated manuscript):
>
> (i) use Python: resulting in 3062 records, the remaining ones span other different languages such as: C (7723), C++ (2860), Java (2685), JavaScript (2292), ruby (1603), rust (1438), etc.
>
> (ii) Filter out the commits that do not modify the test suite, because those would not contain security tests that can detect the fixed vulnerabilities, resulting in 646 records.
>
> (iii) Use Python ≥ 3.7 to avoid vulnerabilities tied to outdated versions and tooling dependencies: resulting in ~450 records.
>
> (vi) fail in execution environment validation: after building the execution environment, we validate the setup by (a) the test cases for security should fail under vulnerable feature implementation and pass under secure implementation; (b) tests for functionality must fail when the feature is masked out. We filtered out all task candidates that did not satisfy these conditions, resulting in 200 tasks
>
> To conclude, we filter out records by (i)(ii)(iii) before we apply our automatic pipelines because we mainly focus on Python repositories with human annotated security tests in this paper. We would like to leave other languages (such as C, Java, etc.) and more human annotations as our future work.
>
> Since we focus on a realistic, high-quality, security-focused benchmark for vibe coding to demonstrate severe security risks in the frontier software engineering agents, we prioritize quality over size. The success ratio of 44.4% shows a balance between quality and size.
>
> ---
>
> **Q1.Could the authors further explain how this example relates to 'spam emails or junk messages'?**
>
> A1: We appreciate the opportunity to clarify this point. In the Django example, the core issue is that the insecure verify_password() implementation introduces a timing side channel that lets an attacker distinguish between existing and non-existing usernames. In many real deployments, usernames are either email addresses or can be trivially mapped to email accounts (e.g., `username@example.com`).
>
> Once an attacker can enumerate which usernames are valid, they can then harvest a high-confidence list of real user accounts, and use this list as input to large-scale spam, junk, or phishing campaigns, credential-stuffing attacks, or targeted account-takeover attempts. Thus, “spam emails or junk messages” was meant as a concrete downstream consequence of username enumeration enabled by the timing leak. Since Django is a widely used web framework in production deployments, this kind of vulnerability directly impacts many real-world services. We also added more explanation of this in our revision at line #403.
>
> ---
>
> **Q2. Could the authors provide more task examples (like Figure 6) or release the entire dataset? Having access to more examples would help better assess the quality of the benchmark.**
>
> A2: We append in-depth analysis of more task cases in SusVibes along with the agent-implemented insecure solution. These are shown at line # 1038 in the appendix of our revision.
>
> ---
>
> **Q3. The authors claim the benchmark consists of 100% cross-file edits. Could they explain why the Django example is considered a cross-file edit, given that it appears to modify only one function in a single file?**
>
> A3: We thank the reviewer for catching this typo and apologize for the confusion. The statement that SusVibes consists of “100% cross-file edits” was not strictly correct. It was a last minute mistake and was the result of miscommunication between our internal authors. SUSVIBES is a repository-level benchmark, but not every task requires modifying multiple files, all existing repository-level coding benchmarks (SWE-Bench, Multi-SWE-Bench, etc.) don't. As we report in Table 2 of our submission, the *golden* solutions for SUSVIBES tasks modify on average 1.8 files and 181.6 lines of code with a maximum of 10 files. We will correct the “100%” claim in the version immediately and rephrase it to emphasize the *average* number of files/lines touched and the repository-level nature of the tasks.
>
> ---
>
> Reference:
>
> [1] SWE-bench: Can Language Models Resolve Real-World GitHub Issues?
>
> [2] Multi-SWE-bench: A Multilingual Benchmark for Issue Resolving

---

### Official Review · Reviewer_9B9S · 2025-11-03

**Soundness:** 3
**Presentation:** 3
**Contribution:** 3
**Rating:** 6
**Confidence:** 3

**Summary:**

This paper proposes SUSVIBES, a novel benchmark that evaluates the security of "vibe coding" by asking agents to complete repository-level tasks. The authors propose a 3-stage agent pipeline for task creation by masking a feature and auto-generating a natural-language "feature request".  The benchmark is grounded in two sets of human-written unit tests, one for correctness and the other for security. The findings reveal that agents are functionally capable but highly insecure. The authors also show that simple prompting-based mitigations fail, often worsening the overall success rate.

**Strengths:**

- The paper addresses a critical, timely, and practical problem, as "vibe coding" represents a rapidly growing phenomenon.
- This work introduces an LLM-assisted pipeline for constructing large-scale data to quantify security risks. The repository-level context, 100% cross-file-edit requirement, and dynamic execution environment present practical and complex challenges. The inclusion of human-written unit tests provides reliability for the "ground truth” of correctness and security.
- The core finding, demonstrating that functional correctness and security are frequently in opposition for LLM agents, represents an important insight. The subsequent findings that simple mitigation strategies (such as prompting) fail and even exacerbate the overall success rate (Figure 5) constitutes a contribution.

**Weaknesses:**

- Curation Pipeline Reliability: Given the finding that current frontier LLM agents perform poorly at implementing security, how do the authors ensure reliability when using these agents for security-critical benchmark construction? Specifically:
  - Can LLM agents generate a "Patch-enclosing feature mask" that accurately captures the necessary security context without over- or under-masking?
  - As the checker in "Security Implication Verifier" (Stage III) is also an LLM agent, it might suffer from the same security blind spots as the agents being evaluated.

More clarification on the reliability of this agent-driven curation process, especially regarding the security implication verification, would strengthen the paper.

- Evaluation Proxy is Narrow: The benchmark's definition of "secure" is to pass the specific human-written tests  that were added to fix the known vulnerability. However, a solution could pass this one test while introducing new vulnerabilities that are not tested for.  This limitation is compounded by the small average of 4.1 security test cases in the dataset.  More evaluation methods such as static and dynamic analysis can be used for vulnerabilities detection.

- Potential Source Data Bias: The benchmark is derived from publicly-identified, fixed, and test-covered vulnerabilities. The benchmark may not represent the full landscape of security issues. The vibe coding agents might be introducing more subtle, novel, or unknown flaws that have no historical precedent and are not in datasets like  (Wang et al., 2024; Akhoundali et al., 2024).

- Missing discussion of some related work, e.g., [1,2,3]

Reference:
- [1] RedCode: Risky Code Execution and Generation Benchmark for Code Agents, NeurIPS 2024 Datasets and Benchmarks Track
- [2] CodeLMSec Benchmark: Systematically Evaluating and Finding Security Vulnerabilities in Black-Box Code Language Models, SATML 2024
- [3] Purple Llama CyberSecEval: A Secure Coding Benchmark for Language Models, 2023

**Questions:**

- Models used for dataset construction: In Section 3.1.1, the authors utilize a 3-stage agent-flow pipeline (Masking, Problem Generation, Verification) to construct the benchmark tasks. Could you please clarify which models/agents were used for these stages?
- CWE Assignment: The paper mentions the dataset is built from existing datasets (Wang et al., 2024; Akhoundali et al., 2024). Could the authors please clarify the specific process used to assign ground-truth CWE categories to each task in the SUSVIBES dataset? For instance, was this an automated mapping from the source datasets, or did it involve manual verification or annotation?

- CWEs per Task: Please clarify if each SUSVIBES task is linked to a single or multiple CWEs. Including statistics on the distribution (e.g., average CWEs per task, percentage of tasks with multiple CWEs) would help in understanding the complexity of the security challenges within the benchmark.

---

> ### Author Response · Authors · 2025-11-25
>
> We thank the reviewer for the valuable and constructive suggestions, which can help us strengthen our work. We are glad to see that the reviewer thinks our vibe coding problem is critical and timely, and agrees that our findings can provide important insights for the security issue in vibe coding. We appreciate the missing related work and will discuss it in our manuscript.
>
> **W1: Curation Pipeline Reliability:**
>
> - **Can LLM agents generate a "Patch-enclosing feature mask" that accurately captures the necessary security context without over- or under-masking?**
> - **As the checker in "Security Implication Verifier" (Stage III) is also an LLM agent, it might suffer from the same security blind spots as the agents being evaluated.**
>
> A1: We appreciate the concern about using LLM agents in the curation pipeline and elaborate on it more in our revision from line #192 to line #247.
>
> To ensure the mask can appropriately cover the feature implementation and the security fixes, we design an iterative masking pipeline with verification. It includes:
>
> (i) An agent to generate an initial mask $\mathcal{M}$. This mask $\mathcal{M}$ is generated on the vulnerable commit before the security fix, i.e., masking out a feature $\mathcal{F}$ from its vulnerable implementation.
>
> (ii) A task description is generated to describe the functionality of this masked implementation
>
> (iii) A verifier agent is used to check whether the task description covers all lines in feature implementation $\mathcal{F}$ + security fixes. Specifically, the agent is prompted to match each line in the secure implementation to a requirement in the description. If any line in the secure implementation is not mentioned by the task description, it will go back to step (i) to regenerate a larger mask; otherwise, go to step (iv)
>
> (iv) Return the task description and the mask that matches the secure/golden implementation
>
> To conclude, instead of solely relying on one agent to do verification, this curated pipeline can ensure that the masked feature and task description cover the feature implementation $\mathcal{F}$ + security fixes. This supervision can mitigate the blind spots of LLM agents in verification
>
> Furthermore, we conduct human evaluation on a randomly sampled subset of 15 tasks (7%) to manually verify the quality. These 15 tasks span across 12 different GitHub projects, accessing 14 different CWEs, forming a representative subset of SusVibes.
>
> 3 software engineering annotators assess the quality of the mask generated by the LLM based on the following criteria:
>
> 1. **Valid Mask:** the mask purely deletes an implementation (or adds at most some place holders)
> 2. **Sufficient Mask:** The mask removes a sufficient feature implementation surrounding the security fix, containing enough security context.
>     1. If the mask covers all lines that were touched by the security fix patch.
>     2. If the mask covers the implementation of a feature that requires the security fix.
> 3. **Necessary Mask:** The mask avoids obviously unrelated or excessive deletions.
>
> The resulting table is shown below.
>
> | Project | CWE Examined | Lines Deleted | Mask is Valid | Mask is Sufficient | Mask is Necessary |
> | --- | --- | --- | --- | --- | --- |
> | `aiohttp` | CWE-444 | 63 | ✓ | ✓ | ✓ |
> | `airflow` | CWE-78 | 83 | ✓ | ✓ | ✓ |
> | `airflow` | CWE-20 | 60 | ✓ | ✓ | ✓ |
> | `airflow` | CWE-79 | 36 | ✓ | ✓ | ✓ |
> | `ckan` | CWE-344 | 115 | ✓ | ✓ | ✓ |
> | `django` | CWE-770 | 225 | ✓ | ✓ | ✓ |
> | `django` | CWE-1333 | 176 | ✓ | ✓ | ✓ |
> | `pandas-ai` | NVD-CWE-noinfo | 151 | ✓ | ✓ | ✓ |
> | `rdiffweb` | CWE-1021 | 106 | ✓ | ✓ | ✓ |
> | `flask` | CWE-539 | 45 | ✓ | ✓ | ✓ |
> | `paramiko` | CWE-362 | 156 | ✓ | ✓ | ✓ |
> | `plone.namedfile` | CWE-79 | 349 | ✓ | ✓ | ✓ |
> | `salt` | CWE-22 | 150 | ✓ | ✓ | ✓ |
> | `vyper` | CWE-787 | 299 | ✓ | ✓ | ✓ |
> | `django-mfa3` | CWE-287 | 169 | ✓ | ✓ | ✓ |
> | *12 projects* | *14 CWEs* | *145.5 lines* | *100%* | *100%* | *100%* |
>
> It is suggested by human software engineers that the masks constructed from the generation- verification pipeline in SusVibes are reasonable in forming security-oriented coding tasks.

---

> ### Author Response · Authors · 2025-11-25
>
> **W2.1 Evaluation Proxy is Narrow: More evaluation methods such as static and dynamic analysis can be used for vulnerability detection.**
>
> A2.1: As suggested by the reviewer, we conduct static analysis on the generated solutions to detect the potential new vulnerabilities that coding agents might introduce.. We followed the static analysis tool used in Asleep[1], and used CodeQL 2.23.5 with an analysis query specifically designed for detecting python vulnerabilities: python-security-quality.qls.
>
> In our experiments, we randomly choose a subset of 20 software engineering tasks in SusVibes. They are examining 12 different CWEs and sourced from 18 real-world GitHub repositories, which forms a representative set of SusVibes’ tasks. We evaluate the SWE-agent’s solutions and human security implementation via CodeQL, and compare the results to see whether SWE-agent introduces new vulnerabilities that humans avoid.
>
> In all tasks, the agent did not introduce critical new vulnerabilities than the human implementation. This indicates that the current code agents can easily avoid vulnerabilities that can be detected by the static analysis. We will add the full results in our further revision.
>
> We agree that the code agents may introduce new vulnerabilities that humans avoid, which may further lower the functionality and security pass ratio we obtain in our paper. If that is the case, it will further strengthen our findings that current code agents perform very poorly in security. On the other hand, we are actively searching for tools to assess the vulnerabilities without human annotations, and find that it is difficult for these tools (as shown above) to detect vulnerabilities in code agent’s solutions. Although SusVibes have covered 77 CWEs, which outperforms the existing benchmarks such as CWEval, BaxBench, SecCodePLT, and SALLM), we believe it can be further enhanced with more human annotated security test cases. We would like to leave this promising direction as our future work.
>
> ---
>
> **W2.2 Potential Source Data Bias: The vibe coding agents might be introducing more subtle, novel, or unknown flaws that have no historical precedent and are not in datasets like (Wang et al., 2024; Akhoundali et al., 2024).**
>
> A2.2: We agree that our benchmark cannot cover all potential security vulnerabilities that vibe coding may introduce. However, our benchmark provides a repository-level testbed for the security issue in vibe coding and covers more vulnerabilities than existing secure coding benchmarks (over 2x more than recently works BaxBench, SecCodePLT, Asleep; 7x more than contemporaneous work SecureAgentBench), as shown in Table 1. SusVibes also cover vulnerabilities that cannot be classified into any existing categories (2% of the tasks), and demonstrate extendability that once novel, publicly recorded vulnerabilities are discovered they can be easily adapted into the benchmark by tracing back to the vulnerable commit and synthesizing the feature request and the runtime evaluation environment.
>
> ---
>
> **W3 related work, [1] RedCode: Risky Code Execution and Generation Benchmark for Code Agents, NeurIPS 2024 Datasets and Benchmarks Track; [2] CodeLMSec Benchmark: Systematically Evaluating and Finding Security Vulnerabilities in Black-Box Code Language Models, SATML 2024; [3] Purple Llama CyberSecEval: A Secure Coding Benchmark for Language Models, 2023**
>
> A3: We appreciate the pointers to RedCode, CodeLMSec, and CyberSecEval and will discuss them in our related work. Here, we demonstrate the difference between these related works and our SusVibes.
>
> - **RedCode** is a safety benchmark for code agents that red-teams them with explicitly risky prompts, evaluating whether they execute or generate harmful code under adversarial instructions.
> - **CodeLMSec** targets black-box code language models rather than agents, automatically mining prompts that cause models to emit vulnerable snippets and checking them via static analysis. The resulting benchmark is a collection of “non-secure prompts” for function-level code generation.
> - **CyberSecEval** benchmarks LLMs’ cybersecurity risks, containing two parts: their propensity to generate insecure code, and their willingness to assist in cyberattacks. We discussed a newer work SecCodePLT, that also lies in this domain in our original manuscript line #145.
>
> Our work, **SusVibes**, focuses on unintentional vulnerabilities introduced when code agents tackle ordinary software-engineering feature tasks on large, real repositories, with correctness and security grounded in real vulnerability-fixing commits and their tests, rather than in red-team or synthetic prompts.

---

> ### Author Response · Authors · 2025-11-25
>
> **Q1. Models used for dataset construction: Could you please clarify which models/agents were used for these stages?**
>
> A1: SWE-agent (v1.1.0) + Claude 4 Sonnet
>
> ---
>
> **Q2. CWE Assignment**
>
> A2: In SusVibes, a task is derived from a vulnerability instance in ReposVul andor Morefixes, and every such instance is linked to an official CVE (Common Vulnerabilities and Exposures) identifier, i.e., a standardized ID for a real-world vulnerability. For each CVE, the ground-truth CWE category is obtained from the upstream datasets directly, which is in turn manually mapped by human annotators in National Vulnerability Database (NVD). We added this in our revision line #1213.
>
> ---
>
> **Q3. CWEs per Task**
>
> A3: We thank the reviewer for pointing this out and will report the CWE distribution more clearly in the revision. Most tasks in SusVibes (97.5%) correspond to a single CWE, while only 3.5% corresponds to multiple CWEs and the maximum number of CWEs each task corresponds to is 2. We will include summary statistics (average CWEs per task and the fraction of single- vs. multi-CWE tasks) and a short table in the appendix.
>
> Although most tasks are anchored to a single CWE, the security challenge is non-trivial: each task is defined at the repository level, requiring the agent to reason over large, multi-file contexts and to implement a security-critical feature that avoids underlying vulnerability. Evaluation results on these tasks reveal serious security concerns in frontier coding agents (over 80% correct solutions are insecure). Our design prioritizes high-fidelity, realistic vulnerabilities over inflating the number of CWE types examined in each individual task.
>
> **Reference:**
>
> [1] Asleep at the Keyboard? Assessing the Security of GitHub Copilot's Code Contributions
>
> [2] RedCode: Risky Code Execution and Generation Benchmark for Code Agents, NeurIPS 2024 Datasets and Benchmarks Track
>
> [3] CodeLMSec Benchmark: Systematically Evaluating and Finding Security Vulnerabilities in Black-Box Code Language Models
>
> [4] SecCodePLT: A Unified Platform for Evaluating the Security of Code GenAI
>
> [5] Baxbench: Can LLMs Generate Correct and Secure Backends

---

### Author Response · Authors · 2025-12-03
**Summary comment for the discussion phase**

We sincerely appreciate the reviewers’ constructive feedback and thoughtful suggestions during the review process. We are glad that the reviewers recognize the importance and timeliness of this work on vibe coding security, and acknowledge SusVibes as a meaningful step toward understanding the risks of agent-generated code in real-world settings.

---

We appreciate that the reviewers acknowledge that:

1. Vibe coding security is a critical, timely research problem (Reviewers **9B9S, 3D9P**)

2. SusVibes advances secure-coding evaluation to repository-level tasks with realistic multi-file editing (Reviewers **DRLX, b85Q, 3D9P**)

3. Our automated curation pipeline is well-motivated and grounded in real vulnerability-fixing commits spanning diverse vulnerability types (Reviewers **9B9S, b85Q**)

4. The study reveals serious security issues and gap between functional correctness and security in frontier code agents, and they are not easily mitigatable (Reviewers **9B9S, 3D9P, b85Q**)

---

We have systematically addressed all concerns raised during the discussion:

1. Elaborate clearly on our curation pipeline with reliability analysis, including a detailed, step-by-step explanation of the iterative mask–verification process, supported by line-level matching of secure implementations **(added in Section 3.1).** (Reviewers **9B9S, DRLX, b85Q**)

2. Conducted manual evaluation of **15 representative tasks spanning 12 repos and 14 CWEs** to validate mask quality in three aspects: validity, sufficiency, necessity.  ****(Reviewers **9B9S, DRLX, b85Q**)

3. Demonstrated that tasks in SusVibes are neither trivial nor homogeneous, but **span a broad difficulty spectrum** with promising discriminative power.  ****(Reviewer **3D9P**)

4. Added **static analysis** to assess whether agents introduce new, subtle vulnerabilities, reinforcing the robustness of evaluation with human-written tests. (Reviewer **9B9S, 3D9P**)

5. Provided real example of the full 3-stage curation pipeline and additional case studies on severe risks coding agents can introduce in **Appendix C.** (Reviewers **b85Q, DRLX**)

6. Clarified provenance and rigorousness of tests, showing that all functionality and security tests are **human-written tests from real repositories**, validated via execution across vulnerable, masked, and fixed commits. (Reviewers **b85Q, DRLX**)

7. Made clear our releasing plan and the further, more comprehensive evaluation with the community's effort. (Reviewer **3D9P**)

---

Moreover, we will enhance our manuscript based on reviewer suggestions (highlighted in green in the revised manuscript)

1. Added clearly presented curation pipeline with details and figures in **Section 3.1 (line #190 - #247)**

2. Expanded discussion of dataset filtering, clearly explaining how 20k records narrow to 200 high-quality, Python ≥3.7, test-covered, reproducible tasks (Reviewer **DRLX**)

3. Enhanced details on the rigorous validation of tests. **(line #260 - #265)**

4. Include a detailed explanation of the case studies in **Section 4.3 and Appendix C**, showing how agents' security risks may relate to your spam emails and junk messages. (Reviewer **DRLX**)

5. Added CWE assignment details, including NVD-based mapping from upstream datasets and statistics on single- vs. multi-CWE tasks in **Appendix D**. (Reviewers **9B9S, 3D9P, b85Q**)

6. Fixed typos and reorganized sections within page limits.

Due to the ICLR incident, we do not get the chance to hear back from the reviewers. We sincerely appreciate the reviewers’ efforts and hope our rebuttal and revisions can be considered during the final decision.

---

### Meta-Review · Area_Chair_PNp8 · 2026-01-05

**Summary:**

The strengths mentioned by the reviewers:
- The problem the paper focuses on is critical and very timely.
- The repository-level context and runtime is more realistic and highly practical.
- The paper finds that frequently there appears to be a trade off between functional correctness and security.
- The authors present a novel and automatic pipeline for constructing tasks, which can be valuable for the community.
- The evaluation provides strong evidence that current coding agents produce insecure code.
- Coverage: The dataset spans 77 CWE categories, capturing diverse vulnerability types across real-world projects.

The weaknesses mentioned by the reviewers:
- How do the authors ensure reliability when using agents to construct the security-critical benchmark when said agents are insecure/unreliable? Do the blind spots from the LLM generating insecure code transfer to the llm that does the security implication verification? **Partially addressed.**
- Can LLM agents generate accurate masks? **Partially addressed.** It appears that not all features can be handled this way. Integrate multi-file edits where code may be modified but not added / changed can potentially not be handled.
- The human written tests are limited and could frequently lead to false negatives. What about static or dynamic analysis? **Partially addressed.** The authors did not observe open source tools finding security issues, although they might have suffered also from false negatives.
- The data is drawn from public repos with publicly-identified fixed and test-covered vulnerabilities. This distribution may be different to the one seen in vibe coding. **Partially addressed.** The synthesized natural language task might be from a different distribution compared to standard vibe-coded queries.
- Some related work is missing. **Addressed**
- The benchmark only includes 200 tasks, despite the authors claiming to have collected ~20,000 open-source vulnerability records over the last 10 years. This raises concerns about the success rate of the pipeline. **Addressed**
- The data distribution is not balanced. **Partially addressed.**
- Insufficient mitigation approach: This work has insights into how insecure the current vibe coding is, while proposing some potential mitigation approaches beyond prompt based techniques will make this paper more solid. **Not addressed.** This is also somewhat out of scope as this paper focuses on creating a benchmark, not on making coding agents more secure.
- It remains unclear how other model families, for example, Qwen, GPT etc. performs on these tasks. **Partially addressed.** No numbers provided for other (perhaps also more affordable) models.
- CWE scope ambiguity: The paper does not make clear whether each task corresponds to a single CWE or involves multiple overlapping CWE types.

The questions asked by the reviewers:
- Models used for dataset construction: In Section 3.1.1, the authors utilize a 3-stage agent-flow pipeline (Masking, Problem Generation, Verification) to construct the benchmark tasks. Could you please clarify which models/agents were used for these stages? **Addressed.**
- CWE Assignment: The paper mentions the dataset is built from existing datasets (Wang et al., 2024; Akhoundali et al., 2024). Could the authors please clarify the specific process used to assign ground-truth CWE categories to each task in the SUSVIBES dataset? For instance, was this an automated mapping from the source datasets, or did it involve manual verification or annotation? **Addressed.**
- CWEs per Task: Please clarify if each SUSVIBES task is linked to a single or multiple CWEs. Including statistics on the distribution (e.g., average CWEs per task, percentage of tasks with multiple CWEs) would help in understanding the complexity of the security challenges within the benchmark. **Addressed.**
- How does the automatic pipeline determine whether a test is related to security, even if it appears in the same commit as the patch? **Addressed.**
- Since the entire problem is generated by LLM, how do the authors justify that each task is reasonable and not vague or missing critical information? For example, could a senior human software engineer complete the task and produce secure code with the provided context? **Partially Addressed.**
- Can the authors further explain how the example relates to 'spam emails or junk messages'? **Addressed.**
- Could the authors provide more task examples (like Figure 6) or release the entire dataset? Having access to more examples would help better assess the quality of the benchmark. **Addressed.**
- The authors claim the benchmark consists of 100% cross-file edits. Could they explain why the Django example is considered a cross-file edit, given that it appears to modify only one function in a single file? **Addressed.**
- Can you please justify why swe-agent was chosen to generate the data? Have you tested other agent systems? How good are they? **Addressed**
- Are there plans to release a leaderboard or ongoing benchmark updates as new vulnerabilities or projects emerge? **Addressed.**
- How do you ensure the difficulty distribution across 200 tasks is reasonable? Could some tasks be too trivial or too difficult, affecting the benchmark's discriminative power? **Partially addressed**. It remains unclear if not a significant portion is too hard.
- Which CWE types are inherently harder to detect or fix? What are the root causes of these differences. **Addressed.**
- Can the authors provide a concrete case study demonstrating the three-stage pipeline process (Stage 1–3) on a real vulnerability example? **Partially addressed.** One or two additional examples in the appendix would have been nice.
- Does each task map to a single CWE, or can it belong to multiple CWEs simultaneously? **Addressed.**

**Reviewer Concerns:**

See above.

**Reviewer Scores:**

- Reviewer 9B9S: $6 \to 6$.
- Reviewer DRLX: $4 \to 6$.
- Reviewer 3D9P: $4 \to 4$.
- Reviewer b85Q: $4 \to 6$.

---

### Decision · Program_Chairs · 2026-01-26

Reject